# Macrolide Resistance in *Bordetella pertussis*: Current Situation and Future Challenges

**DOI:** 10.3390/antibiotics11111570

**Published:** 2022-11-07

**Authors:** Lauri Ivaska, Alex-Mikael Barkoff, Jussi Mertsola, Qiushui He

**Affiliations:** 1Department of Paediatrics and Adolescent Medicine, Turku University Hospital and University of Turku, 20521 Turku, Finland; 2InFLAMES Research Flagship Center, University of Turku, 20520 Turku, Finland; 3Institute of Biomedicine, Centre for Infections and Immunity, University of Turku, 20520 Turku, Finland

**Keywords:** *Bordetella pertussis*, pertussis, whooping cough, macrolides, macrolide resistance, erythromycin, azithromycin, clarithromycin

## Abstract

Pertussis is a highly contagious respiratory infection caused by *Bordetella pertussis* bacterium. The mainstay of treatment is macrolide antibiotics that reduce transmissibility, shorten the duration of symptoms and decrease mortality in infants. Recently, the macrolide resistance of *B. pertussis* has been reported globally but is especially widespread in mainland China. In this review, we aim to summarise the current understanding of the epidemiology, resistance mechanisms and clinical implications of *B. pertussis* macrolide resistance. Since the first appearance of macrolide-resistant *B. pertussis* in Arizona, USA, in 1994, only sporadic cases have been reported outside China. In certain parts of China, on the other hand, up to 70–100% of the recent clinical isolates have been found to be macrolide resistant. Reasons for macrolide resistance being centred upon China during the last decade can only be speculated on, but the dominant *B. pertussis* lineage is different between China and most of the high-income countries. It seems evident that efforts to increase awareness, guide molecular epidemiological surveillance and carry out systematic screening of *B. pertussis* positive samples for macrolide resistance should be implemented globally. In addition, practices to improve the clinical care of infants with pertussis caused by resistant strains should be studied vigorously.

## 1. Introduction

Pertussis, or whooping cough, is a highly contagious respiratory infection caused by *Bordetella pertussis*, a small Gram-negative rod bacterium. Despite extensive vaccinations, whooping cough is resurging in many countries including USA, UK and China [1]. The disease can manifest as a severe life-threatening illness, especially in unvaccinated young infants. A cornerstone of the clinical management of infants with recent onset of pertussis infection is, in addition to supportive care, antibiotic management by macrolide antibiotics. Macrolide treatment might ameliorate the disease when started early after infection onset, before the appearance of paroxysmal cough [2].

Macrolides (erythromycin (ERY), clarithromycin (CHL) and azithromycin (AZT)] are the first line antimicrobials used to treat pertussis patients. Several studies have shown their efficacy in vitro, and in clinical settings for clearance of *B. pertussis* [3,4,5,6].

The first *B. pertussis* strain with decreased sensitivity to macrolide antibiotics was detected in Arizona, USA in 1994 [7]. Since then, macrolide resistant *B. pertussis* has been detected in several countries, although it is rare. However, macrolide resistant *B. pertussis* has been increasingly reported in China during past decade, raising the concern of its potential transmission to other regions and countries. In this review, we aim to describe the epidemiological features, main resistance mechanism, issues with rapid diagnostics, and clinical implications of macrolide resistant *B. pertussis*. Search strategy: We searched PubMed and Google Scholar for articles published before 20 October 2022, by use of the terms: “pertussis” AND “macrolide” AND “resistance”, and reference lists of identified studies. Only articles written in English were included. Finally, only the most relevant papers for this review were citated.

## 2. Pertussis Diagnostics

Pertussis diagnostics can be divided into three main approaches: (1) culture, (2) nucleic acid detection (PCR) and (3) serology. Patient age, vaccination history and onset of the symptoms should be considered when choosing the correct diagnostic method [8]. Culture can be performed up to 2 weeks after the symptoms have appeared, before the bacteria is cleared by the immune defence. Specimen from freshly obtained nasopharyngeal (NP) samples should be cultured on Regan-Lowe (RL, charcoal) or Bordet-Gengou (BG, blood) agar containing cephalexin. Suspected *B. pertussis* specific colonies are further cultured on RL/BG agar (without cephalexin), and identified with e.g., slide agglutination test with specific anti-*B. pertussis* and anti-*B. parapertussis* sera or MALDI-TOF [8,9,10]. Specific nucleic acid identification (targeting IS481/*ptxp*) with PCR requires only a small amount of DNA for detection and identification of the bacterium and is therefore far more sensitive than culture. Furthermore, it can be used even three to four weeks after the onset of symptoms. Therefore, PCR-based approaches are more widely used than culture, especially with infants and small children. For school children and adults, serology is commonly used as there is less interference in antibodies induced from previous vaccinations and the only symptom may have been a prolonged cough (>3–4 weeks, culture nor PCR can be used). Serological diagnosis should be made based on the measurement of serum IgG antibodies against pertussis toxin [11]. Furthermore, laboratory confirmation of *B. pertussis* from clinical samples is needed before antimicrobial susceptibility testing (AST) is performed.

## 3. Epidemiology

The first macrolide resistant *B. pertussis* strain was identified in a 2-month-old infant from Yuma, Arizona, US in 1994 [7]. The isolate was highly resistant to erythromycin with a minimum inhibitory concentration (MIC) > 64 µg/mL. However, the origin of this isolate was not known. Breakpoints to detect antimicrobial resistance of clinical *B. pertussis* isolates were not standardized but the reported resistant strains had MICs of >256 µg/mL with erythromycin (ERY) and clarithromycin (CHL) by Etest method suggesting macrolide resistance. Concurrently, seven additional *B. pertussis* isolates from the same area were tested, but macrolide resistance was not detected in these cases. In a review of 47 *B. pertussis* isolates from children in Utah, US, in 1985–1997, one isolate from January 1997 was resistant against erythromycin [12].

Since the first appearance of macrolide-resistant *B. pertussis*, macrolide susceptibility has been tested in thousands of cultured isolates all over the world (Table 1, Figure 1). In a study of 1030 isolates collected from various parts of the the US, five (0.5%) isolates were erythromycin resistant. Four out of five isolates were from Arizona (1994–1995) and one from Georgia (1995). All isolates initially showed the growth inhibition of *B. pertussis* by disc diffusion method, but after 5–7 days of incubation, novel bacterial colonies appeared on the plate inside the growth inhibition area, demonstrating heterogeneous phenotype [13]. In a review of 38 *B. pertussis* isolates from France in 2003, none of them were resistant to erythromycin [14]. However, nine years later in 2012, the first patient in Europe with macrolide-resistant *B. pertussis* was diagnosed in Lyon, France [15]. A three-week-old neonate with severe pertussis was treated repeatedly with macrolides before the detection of the resistant isolate. Of the three serial isolates from the patient, the first two were sensitive, but the third one turned to be resistant, suggesting that the *B. pertussis* isolate acquired the mutation leading to macrolide resistance during the macrolide treatment. Sporadic cases of macrolide-resistant *B. pertussis* isolates were also reported from Iran in 2009 [16].

In Asia, studies from Cambodia, Japan, Taiwan and Vietnam have found some macrolide-resistant *B. pertussis* isolates that seem to be related to resistant strains in mainland China [17,18,19]. In northern Vietnam, of NP swab samples from 184 patients with pertussis diagnosed during 2016–2020, 24 (13.0%) were found to be resistant. In Japan, the first isolation of a macrolide-resistant strain was from a 2-month-old baby in 2018. The MICs of the isolate showed >256 µg/mL for ERY and CHL and >32 µg/mL for AZT. The complete genome sequence of the macrolide resistant *B. pertussis* strain from Japan has been published [20]. It confirms that the isolate has a homogeneous A2047G mutation in each of the three copies of its 23S rRNA gene and that it belongs to the genotype that is common in Chinese macrolide resistant *B. pertussis* isolates. The issue of macrolide-resistant *B. pertussis* is greater and reported in more detail in China than in anywhere else in the world. The first macrolide-resistant isolates from Shandong Province in China were reported in 2011 in two asymptomatic pupils [21]. No macrolide resistance has been detected in historical isolates in China from 2008 or earlier [22,23]. More recent reports show very high prevalence of macrolide resistance among *B. pertussis* isolates in different parts of China (Table 1).

**Table 1 antibiotics-11-01570-t001:** Global frequencies of macrolide-resistant *Bordetella pertussis*.

Country	Region/City	Year	Resistant Isolates Identified (Frequency %)	Reference
Australia	New South Wales, Perth	1971–2010	0/120 (0.0)	[24,25]
Cambodia	Whole country	2017–2020	1/71 (1.4)	[19]
Canada	Ontario	2011–2013	0/275 (0.0)	[26]
China	Xi’an	2012–2020	274/299 (91.6)	[27,28,29,30,31]
	Shandong	2011	2/2 (100.0)	[21]
	Northern	1970–2014 **	91/124 ** (91.9)	[22]
	Shanghai	2016–2017	81/141 (57.5)	[32]
	Zhejiang	2016–2020	271/381 (71.1)	[33,34,35]
	Beijing, Jinan, Nanjing, Shenzhen	2014–2016	292/335 (87.2)	[36]
	Midwest	2012–2015	163/167 (97.6)	[37]
	Whole country	1950–2018	316/388 (81.4)	[23]
	Hunan	2017–2018	27/55 (49.1)	[38]
	Shenzhen	2015–2017	51/105 (48.6)	[39]
	Whole country	2017–2019	265/311 (85.2)	[40]
Czech republic	Whole country	1967–2015	0/135 (0.0)	[41]
Finland	Whole country	2006–2017	0/148 (0.0)	[42]
France	Bordeaux & Lyon	2003 and 2012	1/41 (2.4)	[10,11]
Iran	Whole country	2009–2010	2/11 (18.2)	[16,43]
Italy	Rome	2012–2015	0/18 (0.0)	[44]
Japan	Whole country	2017–2019	1/33 (3.0)	[17,19]
Taiwan	Whole country	2003–2007	2/76 (2.6)	[19,23]
United Kingdom	Whole country	2001–2009	0/582 (0.0)	[45]
United States	Colorado, Maryland, Oklahoma, Wisconsin	1986	0/75 (0.0)	[46]
	Arizona—Yuma County	1994	1/1 (100.0)	[47]
	Utah	1985–1997	1/47 (2.1)	[12]
	Northern California	1998–1999	0/36 (0.0)	[48]
	Phoenix, Oakland *, San Diego	N/A ***	1/48 (2.1)	[49]
	California, New York, Minnesota, Massachusetts, Illinois, Arizona, Georgia	1994–2000	5/1030 **** (0.5)	[13]
	Minnesota	1997–1999	1/8 (12.5)	[50]
Vietnam	Hanoi, Ha Nam, Thai Binh	2016–2020	24/184 (13.0)	[18,19]

* Hill et al. included a control *B. pertussis* strain, resistant to macrolides. This strain has been isolated in Oakland but not officially published elsewhere. ** Divided into three time periods: 1970s, 2000–2008 and 2013–2014. All isolates (N = 25) collected in 1970–2008 were macrolide sensitive. *** N/A = Not available. **** Notified 5 to 7 days after incubation. Four from Arizona, one from Georgia.

Until recently, macrolide resistance in *B. pertussis* in China has been associated almost exclusively with the *ptxP1* lineage of the bacterium [22,27,29,30,31,32,37]. However, a recent cross-sectional study describes two *ptxP3* isolates from eastern China that had acquired the A2047G mutation in their 23S rRNA gene [40]. The *ptxP3* lineage is currently the dominating *B. pertussis* circulating in most of the high-income countries that have switched to acellular pertussis vaccine in recent decades [51,52]. It has been hypothesized that the replacement of the whole-cell pertussis vaccine with co-purified acellular pertussis vaccine in the national immunization programme, the liberal use of macrolides in children with respiratory infections, and high population densities could have contributed to the effective spread of macrolide-resistant *B. pertussis* in China [53]. 

## 4. Mechanisms behind Macrolide Resistance in *B. pertussis*

Macrolide resistance can be caused by three distinct mechanisms. The most common mechanism, including for *B. pertussis*, is the A2047G single nucleotide polymorphism (SNP) in the 23S rRNA gene within the domain V [15,28,50]. This is equal to a SNP in position A2058G in *E. coli* and A2064G in *M. pneumoniae* [54,55]. The A2047G mutation affects the macrolide binding site in the 23S rRNA component of the 50S ribosomal subunit and prevents macrolides to inhibit the peptide elongation [50]. There are three copies of this gene in the *B. pertussis* genome. Bartkus et al. showed that the A2047G SNP can be found in one or more of the copies. They suggested that this mutation needs at least two copies for resistance [50]. However, many studies have shown that in most cases, all three copies are mutated among the macrolide-resistant *B. pertussis* strains [15,27,37]. 

The second possible cause is the acquisition of the ERY-resistant methylase (*erm*) gene, which leads to addition of methyl group in the 23S rRNA to block the ERY binding site [37,50]. However, *B. pertussis* do not possess this gene, which is also shown in a novel study in which 167 clinical isolates were screened to identify the possible inclusion of this gene. However, none of the strains carried such a gene [37]. So far, no studies have found this mechanism to be the cause of macrolide resistance in *B. pertussis*.

The third proposed mechanism is the expression of MexAB-OprM efflux pump (regulated by the *mexAB-oprM* operon), which helps the bacteria to regulate the uptake of macrolides. This mechanism excretes macrolide molecules out of the bacterial cell. The mechanism has been well-described and has been shown to cause resistance against many antimicrobial agents, including macrolides, in *Pseudomonas aeruginosa* [56]. Lately, Fong et al. described the expression of the *mexAB-oprM* operon within macrolide-resistant *Bordetella parapertussis*. Furthermore, they showed upregulation of the *mexAB-oprM* when *B. parapertussis* was grown in 256 mg/mL of ERY. As no other mechanism was found to cause the resistance, they speculated on the potential effect of this mechanism to cause the resistance. However, they also showed that this operon was not functional in *B. pertussis* due to deletions in *mexA* and *oprM* genes [57]. Whether there will be *B. pertussis* with functional *mexAB-oprM* operon remains to be seen.

There have only been two reports (Iran and China) where the A2047G SNP has not been the mechanism behind the macrolide resistance in *B. pertussis* [22,43]. However, these two studies did not perform *erm* gene or *mexAB-oprM* operon identification, and the reason for the resistance remains unknown. In the study by Mirzaei et al., the macrolide-resistant isolate was resistant to ERY/CHL but not to AZT [43]. Therefore, the presence of *erm* could be the cause of the resistance in these studies and would be the first one detected among macrolide-resistant *B. pertussis*.

## 5. Methods to Detect Macrolide Resistant *B. pertussis*

Antimicrobial susceptibility testing can be performed with cultured *B. pertussis* isolates or with *B. pertussis*-specific DNA. The first approaches to studying AST were performed by agar and broth dilution series, where plates and liquid medium were prepared with standardised antimicrobial agent concentrations [58,59]. Later, disk diffusion (DD) and MIC Etests were adapted, which made the testing less time consuming and simpler to carry out [12]. Eventually, the lack of cultures performed, and the possibility for the easy detection of SNPs, led to the DNA-based identification of macrolide-resistant *B. pertussis* [21,50,60]. Here, we briefly describe the AST methods used currently to identify macrolide-resistant *B. pertussis*.

### 5.1. Disk Diffusion and Minimum Inhibition Concentration Methods

To perform DD or MIC testing, a confirmed *B. pertussis* culture is needed. So far, there are no cut-offs for either of the previous methods recommended by EUCAST, and all determinations for sensitivity or resistance are based on notifications from clinical studies. For both DD and MIC testing, bacterial suspension equivalent to McFarland (McF) standard 0.5 is inoculated on selected culture plates. RL and BG agar plates are used in many studies with 0.5 McF [12,42,49,50]. In addition, Muller–Hinton agar supplemented with blood (HMB) have been used, although studies have shown that 8 McF is needed for confluent growth on this medium [42,45]. For DD testing, an antimicrobial agent disk (ERY, AZT, CHL, clindamycin (CLI)) is placed on the plate, and the disk diffusion zone is measured. Results from the studies performing DD vary, and different intervals have been used for plate read-outs. In general, the DD zone for ERY-susceptible strains varies between 37 and 60 mm, whereas for resistant strains, it is 6mm (reflecting the diagonal of the disk) [12,38,49,59,61]. The DD zone is also affected by the incubation time. Longer incubation leads mostly to an increase in the zone diameter [59]. In general, DD tests are no longer that widely used, and there has been criticism over the reliability of this testing method as the results are not reproducible, have low sensitivity and do not correlate with good clinical outcomes [8,38,62].

The MIC testing is more commonly performed than DD tests. After the Etest (slides with increasing antimicrobial agent concentrations) became available and were evaluated, they more or less replaced the plate dilution methods [49]. The method is simple to perform. After the addition of 0.5 McF *B. pertussis* suspension on a culture plate, an Etest slide is added in the middle of the plate. After 2–5 days, sensitivity to the selected antimicrobial agent can be interpreted as the point where bacterial growth is in touch with the test strip. Figure 2 shows antimicrobial susceptibility testing of *B. pertussis* to ERY (Etests for AZT similar to ERY) on RL charcoal agar with inoculation density equivalent to a 0.5 McFarland standard.

For *B. pertussis*, several studies have been performed for MICs against antimicrobials (ERY, AZT, CHL, CLI). These tests have been quite consistent with the findings. For sensitive strains, MIC has varied from <0.016 to 0.25 µg/mL, whereas nearly all resistant strains have MIC >256 µg/mL [28,31,32,38,42,63]. However, one study in Iran described an isolate that was resistant to ERY (128 µg/mL) and CHL (>256 µg/mL) but not to AZT (<0.06 µg/mL). Furthermore, the authors did not identify the A2047G mutation in this strain as previously described [43]. In addition, Hill et al. and Korgenski et al. described the first two identified macrolide-resistant *B. pertussis* in the USA (Arizona and California) to have an MIC of 64 µg/mL for ERY [12,49]. A flow chart of how to identify macrolide-resistant *B. pertussis* is presented in Figure 3. For *B. pertussis* culture-positive samples, the nucleic acid amplification indicated in the flow chart should be also used for rapid identification of possible A2047G mutation of 23S rRNA.

### 5.2. DNA-Based Identification of A2047G Mutation in the 23S rRNA

There are different approaches to detecting the A2047G mutation. One method is based on the amplification of a 521 bp fraction of the 23S rRNA gene by PCR and its cleavage with *BbsI* restriction enzyme. This results in two separate fragments (393 bp and 128 bp) for resistant isolates and one fragment (521 bp) for sensitive isolates when imaged on a gel [15,27,50]. Another option is the Sanger sequencing of the amplification product to detect the specific A2047G SNP [27,36,50]. However, short-read Sanger sequencing cannot differentiate the three copies of the 23S rRNA gene; long-read sequencing is needed to confirm the number of mutations in the three copies [57]. In addition, whole-genome sequencing (WGS) can be used, but so far, no studies are relying on this method as a sole approach to detecting macrolide-resistant *B. pertussis*. In 2015, Wang et al. introduced an allele-specific PCR to detect the A2047G SNP [60]. In this method, specific primers with small modifications are used to produce either one or two bands after amplification when imaged on a gel. Two bands mark resistance and one band susceptibility of the studied *B. pertussis* isolates. Zhang et al. published another approach based on qPCR high-resolution melting analysis (HRMA) [21]. In this method, the A2047G mutation is identified by the difference in the HRMA melting temperatures of the amplified PCR products. To enhance the HRMA difference, DNA samples were spiked with wild-type DNA. However, the method was only performed with extracted DNA from cultured *B. pertussis*, and its usability among DNA extracted from NP samples needs further evaluation. In general, the above-described methods are currently widely used, especially in China, where most of the macrolide-resistant *B. pertussis* isolates have appeared [28,40,53].

## 6. Conclusions and Perspective

Macrolide antibiotics are the mainstay of both the treatment and prevention of pertussis [2]. Traditionally, ERY has been the most-used macrolide to treat pertussis. It has been shown in a randomized controlled trial that 7 days of erythromycin is adequate to eradicate *B. pertussis* from the nasopharynx [64]. More recently, AZT has replaced ERY as the drug of choice for pertussis, due to being as effective, having higher compliance and causing fewer side effects [65]. Early macrolide treatment has shown to be associated with shorter durations of symptoms, shorter periods of being able to transmit and decreased mortality from pertussis in young infants [66,67,68,69]. Macrolides have been recommended as the first-line therapy for all age groups. The second-line treatment option is sulphamethoxazole/trimethoprim (SMZ-TMP), but because of the potentially severe side effects, it is not recommended for the treatment of the youngest infants <2 months of age.

The emergence of macrolide resistance has raised new questions regarding the optimal treatment of young infants with infection caused by macrolide-resistant *B. pertussis*. In vitro, several classes of antibiotics seem to be effective against *B. pertussis*, including SMZ-TMP, levofloxacin, ampicillin, 3rd-generation cephalosporins, gentamicin and piperacillin-tazobactam [17,34,38,40,46,63]. However, no data regarding clinical benefit of these antibiotics in infants with severe pertussis caused by a macrolide-resistant strain exist. Clinical treatment failure with macrolides in patients with pertussis caused by resistant strains has seldom been documented. 

In two novel studies, piperacillin and cefoperazone-sulbactam were shown to be effective for killing *B. pertussis* both in vitro and in vivo, providing good options for alternative treatment in hospitalized infants if an isolate is identified to be macrolide resistant, although their suitability for young infants still needs to be better studied [34,35]. As stated in the study by Hua et al. [33], a controlled clinical trial including more pertussis patients to be treated with single piperacillin, cefoperazone or other antibiotics is scheduled in Zhejiang, China. The use of alternative therapy for pertussis other than macrolide in outpatients needs clinical studies. 

For the future, it is also worth speculating how the use of co-purified acellular pertussis vaccines versus separately purified acellular pertussis vaccines and changes in the overall use of macrolide antibiotics and population density might affect the epidemiology of macrolide-resistant *B. pertussis* and whether these issues could be targeted to combat the spread of resistant strains. Novel vaccines, such as live attenuated nasal vaccine, that would produce more sterilizing mucosal immunity could also help to address the issue of antibiotic resistance in pertussis [70,71].

So far, the only mechanism identified to cause macrolide resistance has been a point mutation at position 2047 (A2047G) in domain V of the 23S rRNA gene of *B. pertussis*. Therefore, simple methods for the rapid identification of this mutation in clinical microbiology laboratories will provide important help for clinicians to use proper antimicrobials for (prophylactic) treatment of patients, especially young infants. These direct typing methods are even more crucial in the future because culture is less and less used for diagnosis of pertussis. The macrolide resistance of *B. pertussis* has not yet been of clinical concern outside mainland China. However, efforts to increase awareness, guide national/international surveillance and implement systematic screening of *B. pertussis*-positive samples are highly recommended. At the same time, practices for the best possible clinical care of infants with pertussis caused by resistant strains should be studied.

## Figures and Tables

**Figure 1 antibiotics-11-01570-f001:**
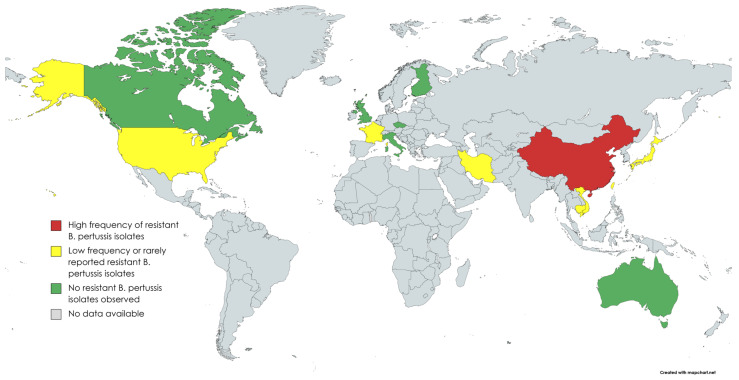
Countries where *B. pertussis* antimicrobial susceptibility studies have been performed (created with MapChart).

**Figure 2 antibiotics-11-01570-f002:**
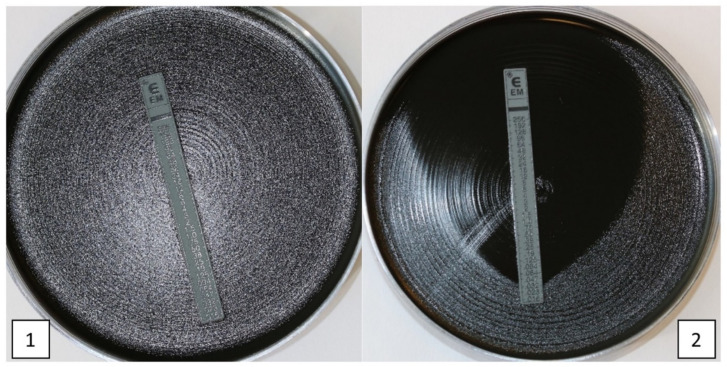
Etest of *B. pertussis* on Regan–Lowe charcoal agar with inoculation density equivalent of 0.5 McFarland standard. (**1**) = erythromycin resistant *B. pertussis* and (**2**) = erythromycin sensitive *B. pertussis*.

**Figure 3 antibiotics-11-01570-f003:**
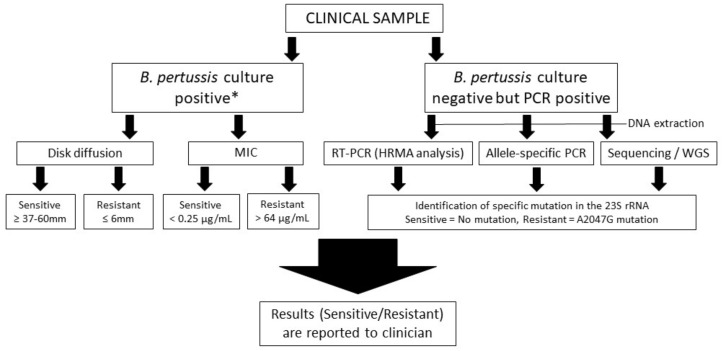
A flow chart of sample processing to detect macrolide-resistant *B. pertussis*. * The A2047G mutation can also be detected from the culture-positive clinical samples by DNA extraction and following the procedure for *B. pertussis* culture-negative but PCR-positive scheme.

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
