# Peer review of "Macrolide Resistance in Bordetella pertussis: Current Situation and Future Challenges"

_antibiotics, 2022, doi:10.3390/antibiotics11111570_

Round 1

Reviewer 1 Report

This manuscript presents a solid perspective on the current Bordetella pertussis pathophysiology and treatment, and the problem that macrolide resistance presents. Nevertheless, I suggest two changes before acceptance:

- More illustrative images should be added, for example in section “5. Methods to detect macrolide resistant B. pertussis”, regarding the described methods; or in section 4, a schematic summarizing the mechanisms behind macrolide resistance in B. pertussis.

- A commentary should be added regarding new strategies to fight macrolide resistance in Bordetella pertussis, namely the most promising new drug molecules under development, and genetic therapies.

Author Response

Responses to reviewer 1 (all page and line numbers refer to the revised version)

Reviewer 1

This manuscript presents a solid perspective on the current Bordetella pertussis pathophysiology and treatment, and the problem that macrolide resistance presents. Nevertheless, I suggest two changes before acceptance:

- More illustrative images should be added, for example in section “5. Methods to detect macrolide resistant B. pertussis”, regarding the described methods; or in section 4, a schematic summarizing the mechanisms behind macrolide resistance in B. pertussis.

Answer: Thank you for the useful comment. We have now added a flow chart showing the process to identify macrolide resistant B. pertussis in section 5

- A commentary should be added regarding new strategies to fight macrolide resistance in Bordetella pertussis, namely the most promising new drug molecules under development, and genetic therapies.

Answer: On page 9, lines 327-330 we report 2 novel studies regarding the use piperacillin and cefoperazone-sulbactam antibiotics against pertussis. We are not aware of any novel antibiotics in clinical trials that would target macrolide resistant B. pertussis. However, a controlled clinical trial including more pertussis patients to be treated with single piperacillin, cefoperazone, or other antibiotics is scheduled in Zhejiang, China. This sentence has been added (page 9, lines 330-333). In addition, we have added a short paragraph (page 9, lines 335-341) addressing potential ways to fight macrolide resistant B. pertussis.

Reviewer 2 Report

The review “Macrolide resistance in Bordetella pertussis: Current situation and future challenges” addresses a very important topic. Overall, the article is well written but there are some suggestions/corrections if addressed will increase the interest of the readers,

1.       Kindly define the B in B. pertussis in the Abstract.

2.       In the abstract it will be hard to follow (ptxP1) & ptxP3) please elaborate.

3.       In the introduction section, please define the abbreviations (ERY, CHL & AZT)

4.       I will suggest the authors include a figure instead of Table 1 or they can show both as a figure showing the world map with the frequency of resistant isolates will increase the understanding of the presented results.

5.       The authors should clearly state the adopted literature search strategy. The readers must know what keywords, mesh terms, or search engines/libraries were used for the literature search and the basis for the inclusion or exclusion of studies. 

Author Response

Responses to reviewer 2 (all page and line numbers refer to the revised version)

The review “Macrolide resistance in Bordetella pertussis: Current situation and future challenges” addresses a very important topic. Overall, the article is well written but there are some suggestions/corrections if addressed will increase the interest of the readers,

  1. Kindly define the B in B. pertussis in the Abstract.

Answer: It has been defined on the first line.

  1. In the abstract it will be hard to follow (ptxP1) & ptxP3) please elaborate.

Answer: We agree. We have thus deleted the two words.

  1. In the introduction section, please define the abbreviations (ERY, CHL & AZT)

Answer: This information has been added.

  1. I will suggest the authors include a figure instead of Table 1 or they can show both as a figure showing the world map with the frequency of resistant isolates will increase the understanding of the presented results.

Answer: Thank you for the important point. We have added a world map (page 5) emphasizing this issue as requested.

  1. The authors should clearly state the adopted literature search strategy. The readers must know what keywords, mesh terms, or search engines/libraries were used for the literature search and the basis for the inclusion or exclusion of studies.

Answer: Thank you for pointing this out. We have added this information to the end of introduction and amended it to show the criteria in better way. We hope that it is now clear for the readers.